# Learning to search efficiently for causally near-optimal treatments

**Samuel Håkansson**[*]
University of Gothenburg
samuel.hakansson@gu.se

**Viktor Lindblom**
Chalmers University of Technology
viklindb@student.chalmers.se

**Omer Gottesman**[†]
Brown University
omer_gottesman@brown.edu

**Fredrik D. Johansson**
Chalmers University of Technology
fredrik.johansson@chalmers.se

## Abstract

Finding an effective medical treatment often requires a search by trial and error. Making this search more efficient by minimizing the number of unnecessary trials could lower both costs and patient suffering. We formalize this problem as learning a policy for finding a near-optimal treatment in a minimum number of trials using a causal inference framework. We give a model-based dynamic programming algorithm which learns from observational data while being robust to unmeasured confounding. To reduce time complexity, we suggest a greedy algorithm which bounds the near-optimality constraint. The methods are evaluated on synthetic and real-world healthcare data and compared to model-free reinforcement learning. We find that our methods compare favorably to the model-free baseline while offering a more transparent trade-off between search time and treatment efficacy.

## 1   Introduction

Finding a good treatment for a patient often involves trying out different options before a satisfactory one is found (Murphy et al., 2007). If the first-line drug is ineffective or has severe side-effects, guidelines may suggest it is replaced by or combined with another drug (Singh et al., 2016). These steps are repeated until an effective combination of drugs is found or all options are exhausted, a process which may span several years (NCCMH, 2010). A long search adds to patient suffering and postpones potential relief. It is therefore critical that this process is made as time-efficient as possible.

We formalize the search for effective treatments as a policy optimization problem in an unknown decision process with finite horizon (Garcia and Ndiaye, 1998). This has applications also outside of medicine: For example, in recommendation systems, we may sequentially propose new products or services to users with the hope of finding one that the user is interested in. Our goal is to perform as few trials as possible until the probability that there are untried actions which are significantly better is small—i.e., *a near-optimal action has been found with high probability*. Historical observations allow us to transfer knowledge and perform this search more efficiently for new subjects. As more actions are tried and their outcomes observed, our certainty about the lack of better alternatives increases. Importantly, even a failed trial may provide information that can guide the search policy.

In this work, we restrict our attention to actions whose outcomes are stationary in time. This implies both that repeated trials of the same action have the same outcome and that past actions do not causally impact the outcome of future actions. The stationarity assumption is justified, for example,

---

[*]This work was completed while the author was affiliated with Chalmers University of Technology.
[†]This work was completed while the author was affiliated with Harvard University.

for medical conditions where treatments manage symptoms but do not alter the disease state itself, or where the impact of sequential treatments is known to be additive. In such settings, past actions and outcomes may help predict the outcomes of future actions without having a causal effect on them.

We formalize learning to search efficiently for causally effective treatments as off-policy optimization of a policy which finds a near-optimal action for new contexts after as few trials as possible. Our setting differs from those typical of reinforcement or bandit learning (Sutton et al., 1998): (i) Solving the problem relies on transfer of knowledge from observational data. (ii) The stopping (near-optimality) criterion depends on a model of unobserved quantities. (iii) The number of trials in a single sequence is bounded by the number of available actions. We address identification of an optimal policy using a causal framework, accounting for potential confounding. We give a dynamic programming algorithm which learns policies that satisfy a transparent constraint on near-optimality for a given level of confidence, and a greedy approximation which satisfies a bound on this constraint. We show that greedy policies are sub-optimal in general, but that there are settings where they return policies with informative guarantees. In experiments, including an application derived from antibiotic resistance tests, our algorithms successfully learn efficient search policies and perform favorably to baselines.

## 2    Related work

Our problem is related to the bandit literature, which studies the search for optimal actions through trial and error (Lattimore and Szepesvári, 2020), and in particular to contextual bandits (Abe et al., 2003; Chu et al., 2011). In our setting, a very small number of actions is evaluated, with the goal of terminating search as early as possible. This is closely related to the fixed-confidence variant of best-arm identification (Lattimore and Szepesvári, 2020, Chapter 33.2), in which only exploration is performed. To solve this problem without trying each action at least once, we rely on transferring knowledge from previous trials. This falls within the scope of transfer and meta learning. Liao et al. (2020) explicitly tackled pooling knowledge across patients data to determine an optimal treatment policy in an RL setting and Maes et al. (2012) devised methods for meta-learning of exploration policies for contextual bandits. A notable difference is that we assume that outcomes of actions are stationary in time. We leverage this both in model identification and policy optimization.

Experiments continue to be the gold standard for evaluating adaptive treatment strategies (Nahum-Shani et al., 2012). However, these are not always feasible due to ethical or practical constraints. We approach our problem as causal estimation from observational data (Rosenbaum et al., 2010; Robins et al., 2000), or equivalently, as off-policy policy optimization and evaluation (Precup, 2000; Kallus and Santacatterina, 2018). Unlike many works, we do not fully rely on ignorability—that all confounders are measured and may be adjusted for. Zhang and Bareinboim (2019) recently studied the non-ignorable setting but allowed for limited online exploration. In this work we aim to bound the effect of unmeasured confounding rather than to eliminate it using experimental evidence.

Our problem is closely related to active learning (Lewis and Gale, 1994), which has been used to develop testing policies that minimize the expected number of tests performed before an underlying hypothesis is identified. For a known distribution of hypotheses, finding an optimal policy is NP-hard (Chakaravarthy et al., 2007), but there exists greedy algorithms with approximation guarantees (Golovin et al., 2010). In our case, (i) the distribution is unknown, and (ii) hypotheses (outcomes) are only partially observed. Our problem is also related to optimal stopping (Jacka, 1991) of processes but differs in that the process is controlled by our decision-making agent.

## 3    Learning to search efficiently for causally near-optimal treatments

We consider learning policies $\pi \in \Pi$ that search over a set of actions $\mathcal{A} \coloneqq \{1, ..., k\}$ to find an action $a \in \mathcal{A}$ such that its outcome $Y(a) \in \mathcal{Y}$ is near-optimal. When such an action is found, the search should be terminated as early as possible using a special stop action, denoted $a = \textsc{stop}$. Throughout, a high outcome is assumed to be preferred and we often refer to actions as "treatments". The potential outcome $Y(a)$ may vary between subjects (contexts) depending on *baseline covariates* $X \in \mathcal{X} \subseteq \mathbb{R}^d$ and unobserved factors. When a search starts, all potential outcomes $\{Y(a) : a \in \mathcal{A}\}$ are unobserved, but are successively revealed as more actions are tried, see the illustration in Figure 1. To guide selection of the next action, we learn from observational data of previous subjects.

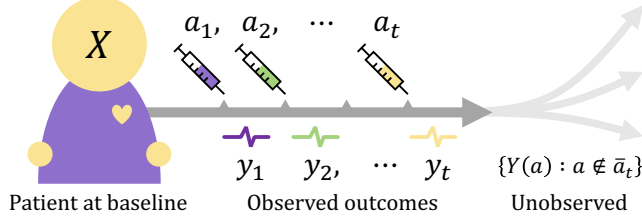

$$a_1, \quad a_2, \quad \cdots \quad a_t$$

$$y_1 \quad y_2 \quad \cdots \quad y_t \qquad \{Y(a) : a \notin \bar{a}_t\}$$

Patient at baseline      Observed outcomes      Unobserved

Figure 1: Illustration of the observed sequence of treatments $\bar{a}_t = (a_1, ..., a_t)$ and outcomes $\bar{y}_t = (y_1, ..., y_t)$ for a patient, and the problem of estimating the outcome of possible future treatments.

Historical searches are observed through covariates $X$ and a sequence of $T$ *action-outcome* pairs $(A_1, Y_1), ..., (A_T, Y_T)$. Note the distinction between interventional and observational outcomes; $Y_s(a)$ represents the *potential* outcome of performing action $a$ at time $s$ (Rubin, 2005). We assume that $y \in \mathcal{Y}$ are discrete, although our results may be generalized to the continuous case. Sequences of $s$ random variables $(A_1, ..., A_s)$ are denoted with a bar and subscript, $\overline{A}_s \in \overline{\mathcal{A}}_s$, and $H_s = (X, \overline{A}_s, \overline{Y}_s) \in \mathcal{H}_s := \mathcal{X} \times \{\overline{\mathcal{A}}_s \times \overline{\mathcal{Y}}_s\}$ denotes the history up-to time $s$, with $H_0 = (X, \emptyset, \emptyset)$. With slight abuse of notation, $a \in h$ means that $a$ was used in $h$ and $(a, y) \in h$ that it had the outcome $y$. $|h|$ denotes the number of trials in $h$. The set of histories of at most $k$ actions is denoted $\mathcal{H} = \cup_{s=1}^{k} \mathcal{H}_s$. Termination of a sequence is indicated by the sequence length $T$ and may either be the result of finding a satisfactory treatment or due to censoring. Hence, the full set of potential outcomes is not observed for most subjects. Observations are distributed according to $p(X, T, \overline{A}_T, \overline{Y}_T)$.

We optimize a deterministic policy $\pi$ which suggests an action $a$ following observed history $h$, starting with $(x, \emptyset, \emptyset)$, or terminates the sequence. Formally, $\pi \in \Pi \subseteq \{\mathcal{H} \to \mathcal{A} \cup \{\text{STOP}\}\}$. Taking the action $\pi(h_s) = \text{STOP}$ at a time point $s$ implies that $T = s$. Let $p_\pi(X, \overline{A}, \overline{Y}, T)$ be the distribution in which actions are drawn according to the policy $\pi$. For a given *slack parameter* $\epsilon \geq 0$ and a *confidence parameter* $\delta \geq 0$, we wish to solve the following problem.

$$
\begin{aligned}
\underset{\pi \in \Pi}{\text{minimize}} \quad & \mathbb{E}_{X, \overline{Y}, \overline{A}, T \sim p_\pi}[T] \\
\text{subject to} \quad & \Pr\left[\max_{a \notin \overline{A}_t} Y(a) > \max_{(a,y) \in h} y + \epsilon \; \middle| \; H_t = h, T = t\right] \leq \delta, \quad \forall t \in \mathbb{N}, h \in \mathcal{H}_t
\end{aligned}
\tag{1}
$$

In (1), the objective equals the expected search length under $\pi$ and the constraint enforces that termination occurs only when there is low probability that a better action will be found among the unused alternatives. Note that if $\max_{y \in \mathcal{Y}} y$ is known and is in $\overline{Y}_s$, the constraint is automatically satisfied at $s$. To evaluate the constraint, we need a model of unobserved potential outcomes. This is dealt with in Section 4. We address optimization of (1) for a known model in Section 5.

## 4   Causal identification and estimation of optimality conditions

Our assumed causal model for observed data is illustrated graphically in Figure 2a. Most notably, the graph defines the causal structure between actions and outcomes—previous actions $A_1, ..., A_{s-1}$ and outcomes $Y_1, ..., Y_{s-1}$ are assumed to have no direct causal effect on future outcomes $Y_s, ..., Y_T$. To allow for correlations between outcomes, we posit the existence of counfounders $X$ (observed) and $U$ (unobserved) and an unobserved moderator $Z$. All other variables are assumed exogenous.

To evaluate the near-optimality constraint and solve (1), we must identify the probability

$$
\rho(h) := \Pr[\max_{a \notin h} Y(a) > \max_{(a,y) \in h} y + \epsilon \mid H = h],
\tag{2}
$$

with the convention that $\max_{(a,y) \in h_0} y = -\infty$ if $|h_0| = 0$. Henceforth, let $\mathcal{A}_{-h} = \{a \in \mathcal{A} : a \notin h_s\}$ denote the set of untried actions at $h$ and let $\mathcal{S}(\mathcal{A}_{-h})$ be all permutations of the elements in $\mathcal{A}_{-h}$.

We state assumptions sufficient for identification of $\rho(h)$ below. Throughout this work we assume that consistency, stationarity of outcomes and positivity always hold, and provide identifiability results both when ignorability holds (Section 4.1) and when it is violated (Section 4.2).

**Identifying assumptions.** *Define $\overline{A}_{s+1:k} = (A_{s+1}, ..., A_k)$. Under the observational distribution $p$, and evaluation distribution $p_\pi$, for all $\pi \in \Pi$, $h \in \mathcal{H}_s$, and $s, r \in \mathbb{N}$, we assume*

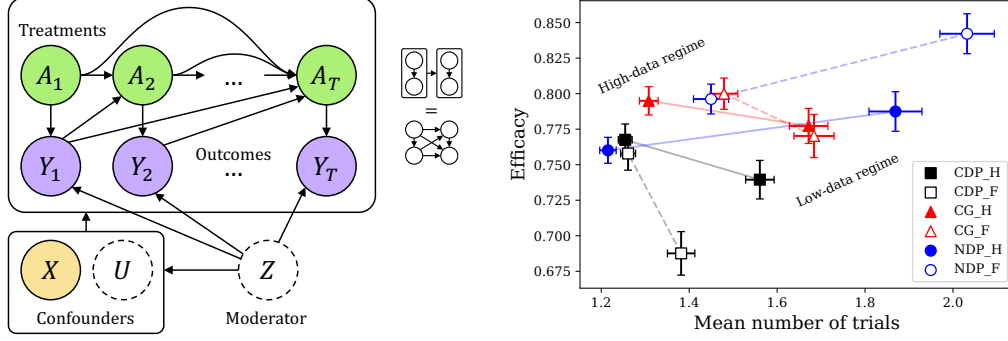

(a) Assumed causal structure. Arrows between boxes indicate connections between *all* variables in the boxes: $X$ is a cause of every $A_s$ and $Y_s$. Past actions are assumed not to be direct causes of future outcomes. $Z$ is a moderator of treatment effects. Dashed outlines indicate unobserved variables.

(b) Efficacy (fraction of subjects for which optimal action is found) and search length for varying amounts of samples, with trade-off parameters $\delta = 0.4, \epsilon = 0$ (CDP, CG), $\lambda = 0.35$ (NDP). The suffix _H indicates historical smoothing and _F function approximation. Error bars indicate standard errors over 71 realizations.

Figure 2: Assumed causal structure (left) and results from synthetic experiments (right).

1. **Consistency:** $Y_s = Y_s(A_s)$

2. **Stationarity:** $Y_s(a) = Y_r(a) =: Y(a)$

3. **Positivity:** $\exists \overline{a} \in \mathcal{S}(\mathcal{A}_{-h_s}) : p_\pi(H_s = h_s) > 0 \implies p(\overline{A}_{s+1:k} = \overline{a} \mid H_s = h_s) > 0$

4. **Ignorability:** $Y_s(a) \perp\!\!\!\perp A_s \mid H_{s-1}$

Ignorability follows from the backdoor criterion applied to the causal model of Figure 2a when $U$ is empty (Pearl, 2009). We expand on this setting next. In contrast to conventions typically used in the literature, positivity is specified w.r.t. the considered policy class. This ensures that every action could be observed *at some point* after every history $h$ that is possible under policies in $\Pi$. Under Assumption 2 (stationarity), there is no need to try the same treatment twice, since the outcome is already determined by the first trial. We can restrict our attention to non-repeating policies,

$$\Pi \subseteq \{\pi \colon \mathcal{H} \to \mathcal{A} \cup \{\textbf{STOP}\} \; ; \; \pi(h) \notin h\} \; .$$

Non-repeating policies such as these take the form of a decision tree of depth at most $k = |\mathcal{A}|$.

**Remark 1** (Assumptions 1–4 in practice). Only the positivity assumption may be verified empirically; stationarity, consistency and ignorability must be justified by domain knowledge. Readers experienced with causal estimation will be familiar with the process of establishing ignorability and consistency through graphical arguments or reasoning about statistical independences. Stationarity is more specific to our setting and without it, the notion of a near-optimal action is not well-defined—the best action could change with time. This phenomenon occurs is settings where outcomes naturally increase or decrease over time, irrespective of interventions. For example, the cognitive function of patients with Alzheimer's disease tends to decrease steadily over time (Arevalo-Rodriguez et al., 2015). As a result, measures of cognitive function $Y_t(a)$ for patients on a medication $a$ will be different depending on the stage $t$ of progression that the patient is in. As a rule-of-thumb, stationarity is better justified over small time-frames or for more stable conditions.

## 4.1 Identification without unmeasured confounders

Our stopping criterion $\rho(h)$ is an interventional quantity which represents the probability that an unused action *would be* preferable to previously tried ones. In general, this is not equal to the rate at which such an action was preferable in observed data. Nevertheless, we prove that $\rho(h)$ is identifiable from observational data in the case that $U$ does not exist (ignorability holds w.r.t. $H$). First, the following lemma shows that the *order* of history does not influence the probability of future outcomes.

**Lemma 1.** *Let $\mathcal{I}$ be a permutation of $(1, ..., s)$. Under stationarity, for all $\overline{a} \in \overline{\mathcal{A}}_s$ and $b \notin \overline{a}$,*

$$p(Y(b) \mid X, \overline{A}_s = \overline{a}, \overline{Y}_s = \overline{y}) = p(Y(b) \mid X, \overline{A}_s = (a_{\mathcal{I}(1)}, ..., a_{\mathcal{I}(s)}), \overline{Y}_s = (y_{\mathcal{I}(1)}, ..., y_{\mathcal{I}(s)})) \quad (3)$$

Lemma 1 is proven in Appendix A.1. As a consequence, we may treat two histories with the same events in different order as equivalent when estimating $p(Y(a) \mid H)$.

We can now state the following result about identification of the near-optimality constraint of (1).

**Theorem 1.** *Under Assumptions 1–4, the stopping criterion $\rho(h)$ in (2) is identifiable from the observational distribution $p(X, T, \overline{A}, \overline{Y})$. For any time step $s$ with history $h_s$, let $h(\mathcal{I})_s = (x, a_{\mathcal{I}(1)}, ..., a_{\mathcal{I}(s)}, y_{\mathcal{I}(1)}, ..., y_{\mathcal{I}(s)})$ be an arbitrary permutation of $h_s$. Then, for any sequence of untried actions $\overline{a}_{s+1:k} = (a_{s+1}, ..., a_k) \in \mathcal{S}(\mathcal{A}_{-h_s})$ with $h(\mathcal{I})_r$ the (hypothetical) continued history at time $r > s$ corresponding to $\overline{a}_{s+1:k}$ and $\overline{y}_{s+1:k}$, and with $\mu(h_s) = \max_{(a,y) \in h_s} y$,*

$$\rho(h_s) = \sum_{\overline{y}_{s+1:k} \in \mathcal{Y}^{k-s}} \mathbb{1}\left[\max(\overline{y}) > \mu(h_s) + \epsilon\right] \prod_{r=s+1}^{k} p(Y_r = y_r \mid A_r = a_r, H_r = h(\mathcal{I})_{r-1}) . \quad (4)$$

A proof of Theorem 1 is given in Appendix A.2. Equation (4) gives a concrete means to estimate $\rho(h)$ from observational data by constructing a model of $p(Y_s \mid A_s, H_{s-1} = h)$. Due to Assumption 2 (stationarity), this model can be invariant to permutations of $h$. Another important consequence of this result is that, because Theorem 1 holds for any future sequence of actions, (4) holds also over any convex combination for different future action sequences, such as the expectation over the empirical distribution. Using likely sequences under the behavior policy will lead to lower-variance estimates.

**Remark 2.** In the fully discrete case, we may estimate $p(Y \mid A, H)$ using a probability table, and we do so in some experiments in Section 6. However, this becomes increasingly difficult for both statistical and computational reasons when $\mathcal{A}$ and $\mathcal{Y}$ grow larger or when any of the variables are continuous. The permutation invariance given by Theorem 1 provides some relief but, nevertheless, the number of possible combinations (histories) grows exponentially with the number of actions. As a result, it is very probable that certain pairs of histories and actions $(h, a)$ are never observed in practical applications. We consider two remedies to this. In Appendix B, we give methods for leveraging observations of similar histories $h' \approx h$ in the estimation of $p(Y \mid H = h, A)$, one based on historical kernel-smoothing in the tabular case, and one based on function approximation. These are compared empirically in Section 6. In Section 5.2, we give bounds to use in place of the probability of unobserved potential outcomes which further mitigate the curse of dimensionality.

## 4.2 Accounting for unobserved confounders

If Assumption 4 (ignorability) does not hold with respect to observed variables, the stopping criterion $\rho(h_s)$ may not be identified from observational data without further assumptions. A natural relaxation of ignorability is that the same condition holds w.r.t. an expanded adjustment set $(H_s, U)$, where $U \in \mathcal{U}$ is an unobserved set of variables. This is the case in our assumed causal model, see Figure 2a. We require additionally that $U$ has bounded influence on treatment propensity. For all $u \in \mathcal{U}, h \in \mathcal{H}$, with $s = |h|$ and $\overline{a} \in \mathcal{S}(\mathcal{A}_{-h})$, assume that there is a sensitivity parameter, $\alpha \geq 1$, such that

$$\frac{1}{\alpha} \leq \frac{\Pr[\overline{A}_{s+1:k} = \overline{a} \mid H_s = h]}{\Pr[\overline{A}_{s+1:k} = \overline{a} \mid U = u, H_s = h]} \leq \alpha , \quad (5)$$

where $\overline{A}_{s+1:k}$ is defined as in Assumption 3. Like ignorability, this assumption must be justified from external knowledge since $U$ is unobserved. We arrive at the following result.

**Theorem 2.** *Assume that (5) and Assumptions 1–4 hold with respect to $(H_s, U)$ for all $s \in [k]$ with sensitivity parameter $\alpha \geq 1$. Then, for any $h \in \mathcal{H}_s, \overline{a} \in \mathcal{S}(\mathcal{A}_{-h})$ and $\nu = \mu(h) + \epsilon$, we have*

$$\Pr[\max_{r=s}^{k} Y_r > \nu \mid \overline{A}_{s+1:k} = \overline{a}, H_s = h] \leq \frac{\delta}{\alpha} \implies \rho(h) = \Pr[\max_{a \in \overline{a}} Y(a) > \nu \mid H_s = h] \leq \delta .$$

A proof of Theorem 2 is given in Appendix A.4. To achieve near-optimality with confidence level of $\delta$ in the presence of unobserved confounding with propensity influence $\alpha$, we must require a confidence level of at most $\delta/\alpha$. Unlike classical approaches to sensitivity analysis, as well as more recent results (Kallus and Zhou, 2018), this argument does not rely on importance (propensity) weighting.

# 5 Policy optimization

We give two algorithms for policy optimization under the assumption that a model of the stopping criterion $\rho(h)$ is known. As noted previously, this problem is NP-hard due to the exponentially increasing number of possible histories (Rivest, 1987). Nevertheless, for moderate numbers of actions, we may solve (1) exactly using dynamic programming, as shown next. Then we propose a greedy approximation algorithm and discuss model-free reinforcement learning as alternatives.

## 5.1 Exact solutions with dynamic programming

Let $X, A, Y$ be discrete. For sufficiently small numbers of actions, we can solve (1) exactly in this setting. Let $h' = h \cup \{(a, y)\}$ denote the history where $(a, y)$ follows $h$ and recall the convention $\max_{a \in \emptyset} Y(a) = -\infty$. Now define $Q$ to be the expected cumulative return—see e.g., Sutton et al. (1998) for an introduction—of taking action $a$ in a state with history $h \in \mathcal{H}$,

$$Q(h, a) = r(h, a) + \mathbb{1}[a \neq \textbf{STOP}] \sum_{y \in \mathcal{Y}} p(Y(a) = y \mid h) \max_{a' \in \mathcal{A} \cup \{\textbf{STOP}\}} Q(h \cup \{(a, y)\}, a'), \quad (6)$$

where $r(h, a)$ is a reward function defined below. The value function $V$ at a history $h$ is defined in the usual way, $V(h) = \max_a Q(h, a)$. To satisfy the near-optimality constraint of (1), we use an estimate of the function $\rho(h)$, see (2), to define $\gamma_{\epsilon, \delta, \alpha}(h) := \mathbb{1}[\rho(h) < \delta/\alpha]$ for parameters $\epsilon, \delta \geq 0$, $\alpha \geq 1$. The function $\gamma_{\epsilon, \delta, \alpha}(h)$ represents whether an $\epsilon, \delta/\alpha$-optimum has been found. We define

$$r_{\epsilon, \delta, \alpha}(h, a) = \begin{cases} -\infty, & \text{if } a = \textbf{STOP}, \gamma_{\epsilon, \delta, \alpha}(h) = 0 \\ 0, & \text{if } a = \textbf{STOP}, \gamma_{\epsilon, \delta, \alpha}(h) = 1 \\ -1, & \text{if } a \neq \textbf{STOP} \end{cases}. \quad (7)$$

With this, given a model of $p(Y_s(a) \mid H_{s-1}, A_s)$, the $Q$-function of (6) may be computed using dynamic programming, analogous to the standard algorithm for discrete-state reinforcement learning.

**Theorem 3.** *Recall that $H_0 = (X, \emptyset, \emptyset)$. The policy maximizing (6), $\pi(h) = \arg\max_a Q(h, a)$, with reward given by (7) is an optimal solution to (1) with objective $\mathbb{E}_{p_\pi}[T] = \mathbb{E}_X[-V(H_0)]$.*

Theorem 3 follows from Bellman optimality and the definition of $r$ in (7), see Appendix A.5.

## 5.2 A greedy approximation algorithm

We propose a greedy policy as an approximate solution to (1) in high-dimensional settings where exact solutions are infeasible to compute. We then discuss sub-optimality and approximation ratios of greedy algorithms. First, consider the greedy policy $\pi_G$, which chooses the treatment with the highest probability of finding a best-so-far outcome, weighted by its value, according to

$$f(h, a) = \mathbb{E}[\mathbb{1}[Y(a) > \max_{(\cdot, y) \in h} y] Y(a) \mid H_s = h] \quad (8)$$

until the stopping criterion is satisfied,

$$\pi_G(h) := \begin{cases} \textbf{STOP}, & \gamma_{\epsilon, \delta, \alpha}(h) = 1 \\ \arg\max_{a \notin h} f(h, a), & \text{otherwise} \end{cases}, \quad (9)$$

where $\gamma_{\epsilon, \delta, \alpha}$ is defined as in Section 5.1. While using $\pi_G$ avoids solving the costly dynamic programming problem of the previous section, it still requires evaluation of $\gamma(h)$. Even for short histories, $|h| \approx 1$, computing $\gamma(h)$ involves modeling the distribution of maximum-length sequences over potentially $|\mathcal{Y}|^{|\mathcal{A}|}$ configurations. To increase efficiency, we bound the stopping statistic $\rho$, and approximate $\gamma$, using conditional distributions of the potential outcome of single actions.

$$\rho(h) := p\left(\max_{a \notin h} Y(a) > \mu(h) + \epsilon \mid h\right) \leq \sum_{a \notin h} p(Y(a) > \mu(h) + \epsilon \mid h). \quad (10)$$

A proof is given in Appendix A.3. Using the upper bound in place of $\rho(h)$ leads to a feasible solution of (1) with more conservative stopping behavior and better outcomes but worse expected search time. In the case $\delta = 0$, the exact statistic and the upper bound lead to identical policies. Representing the upper bound as a function of all possible histories still requires exponential space in the worst case, but only a small subset of histories will be observed for policies that terminate early. We use the bound on $\rho(h)$ in experiments with both dynamic programming and greedy policies in Section 6. The general problem of learning bounds on potential outcomes was studied by (Makar et al., 2020).

**Example 1.** In the following example, the greedy policy does identify a near-optimal action after the smallest expected number of trials, for $\delta = 0, \epsilon = 0$. Let $X = 0, Y \in \{0, 1\}$ and $A \in \{1, 2, 3\}$, $Z = \{1, ..., 4\}, p(Z) = [0.20, 0.15, 0.20, 0.45]^T$ and let $C$ be the matrix with elements $c_{ij}$ such that

$$p(Y(j) = 1 \mid Z = i) = c_{ij}, \text{ with } C^\top = \begin{bmatrix} 1 & 0 & 1 & 0 \\ 0 & 1 & 0 & 1 \\ 1 & 0 & 0 & 1 \end{bmatrix}.$$

In this scenario, $p(Y(\cdot) = 1) = [0.4, 0.6, 0.65]^\top$. The greedy strategy would thus start with $\pi(\emptyset) = 3$, followed by $\pi((3)) = 2$ and then $\pi((3, 2)) = 1$ to guarantee successful treatment. An optimal strategy is to start with $A_1 = 2$ and then $A_2 = 1$. The expected time $\mathbb{E}[T]$ is 1.5 under the greedy policy and 1.4 under the optimal one. The worst-case time under the greedy strategy is 3 and 2 under the optimal.

In Appendix A.6, we show that our problem is equivalent to a variant of active learning once a model for $p(Y(a_1), ..., Y(a_k), X)$ is known. In general, it is NP-hard to obtain an approximation ratio better than a logarithmic factor of the number of possible combinations of potential outcomes (Golovin et al., 2010; Chakaravarthy et al., 2007). However, for instances with additional structure, e.g., through correlations induced by the moderator $Z$, this ratio may be significantly smaller than $|\mathcal{A}| \log |\mathcal{Y}|$.

### 5.3 A model-free approach

In off-policy evaluation, it has been noted that for long-term predictions, model-free approaches may be preferable to, and suffer less bias than, their model-based counterparts (Thomas and Brunskill, 2016). They are therefore natural baselines for solving (1). We construct such a baseline below.

Let $\max_{(\cdot, y) \in h} y$ represent the best outcome so far in history $h$, with $s = |h|$ and let $\lambda > 0$ be a parameter trading off early termination and high outcome. Now, consider a reward function $r(h, a)$ which assigns a reward at termination equal to the best outcome found so far. A penalty $-\lambda$ is awarded for each step of the sequence until termination, a common practice for controlling sequence length in reinforcement learning, see e.g, (Pardo et al., 2018). Let

$$r_\lambda(h, a) = \{0, \text{ if } a \neq \text{STOP } ; \quad \max_{(\cdot, y) \in h} y - \lambda|h|, \text{ if } a = \text{STOP}\} . \tag{11}$$

The policy $\pi_\lambda$ which optimizes this reward, using dynamic programming as in Section 5.1, is used as a baseline in experiments in Section 6. While this approach has the advantage of not requiring a model of future outcomes, *without a model, the stopping criterion $\rho(h)$ cannot be verified and the advantage of being able to specify an interpretable certainty level is lost.* This is because the trade-off parameter $\lambda$ does not have a universal interpretation—the value of $\lambda$ which achieves a given rate of near-optimality will vary between problems. In contrast, the confidence parameter $\delta$ directly represents a bound on the probability that there is a better treatment available when stopping. Additionally, in Appendix A.7, we prove that there are instances of the main problem (1), for a given value of $\delta$, such that no setting of $\lambda$ results in an optimal solution.

## 6 Experiments

We evaluate our proposed methods using synthetic and real-world healthcare data in terms of the quality of the best action found, and the number of trials in the search.[3] In particular, we study the *efficacy* of policies, defined as the fraction of subject for which a near-optimal action has been found when the choice to stop trying treatments is made. Models of potential outcomes are estimated using either a table with historical smoothing (labeled with suffix _H) or using function approximation using random forests (suffix _F), see Appendix B. Following each estimation strategy, we compare policies learned using constrained dynamic programming (CDP), the constrained greedy approximation (CG) and the model-free RL variant, referred to as as naïve dynamic programming (NDP), see Section 5. Establishing near-optimality is infeasible in most observational data as only a subset of actions are explored. However, as we will see, in our particular application, it may be determined exactly.

### 6.1 Synthetic experiments: Effect of sample size and algorithm choice

To investigate the effects of data set size, number of actions, dimensionality of baseline covariates and the uncertainty parameter $\delta$ on the quality of learned policies, we designed a synthetic data generating

process (DGP). This DGP parameterizes probabilities of actions and outcomes as log-linear functions of a permutation-invariant vector representation of history and of $(X, Z)$, respectively. For the results here, $\mathcal{A} = \{1, ..., 5\}, \mathcal{X} = \{0, 1\}, \mathcal{Y} = \{0, 1, 2\}, \mathcal{Z} = \{0, 1\}^3$. Due to space limitations, we give the full DGP and more results of these experiments in Appendix C.1.

We compare the effect of training set size for the different policy optimization algorithms (CDP, CG, NDP and model estimation schemes (_F, _H). Here, CDP and CG use $\delta = 0.4, \epsilon = 0$ and the upper bound of (10) and NDP $\lambda = 0.35$. We consider training sets in a low-data regime with $50$ samples and a high-data regime of $75000$, with fixed test set size of $3000$ samples. Results are averaged over $71$ realizations. In Figure 2b, we see that the value of all algorithms converge to comparable points in the high-data regime but vary significantly in the low-data regime. In particular, CG and CDP improve on both metrics as the training set grows. The time-efficacy trade-off is more sensitive to the amount of data for NDP than for the other algorithms, and while additional data significantly reduces the mean number of actions taken, this comes at a small expense in terms of efficacy. This highlights the sensitivity of the naïve RL-based approach to the choice of reward: the scale of the parameter $\lambda$ determines a trade-off between the number of trials and efficacy, the nature of which is not known in advance. In contrast, CDP and CG are preferable in that $\delta$ and $\epsilon$ have explicit meaning irrespective of the sample and result in a subject-specific stopping criterion, rather than an average-case one.

## 6.2 Optimizing search for effective antibiotics

Antibiotics are the standard treatment for bacterial infections. However, infectious organisms can develop resistance to specific drugs (Spellberg et al., 2008) and patterns in organism-drug resistance vary over time (Kanjilal et al., 2018). Therefore, when treating patients, it is important that an antibiotic is selected to which the organism is susceptible. For conditions like sepsis, it is critical that an effective antibiotic is found within hours of diagnosis (Dellinger et al., 2013).

As a proof-of-concept, we consider the task of selecting effective antibiotics by analyzing a cohort of intensive-care-unit (ICU) patients from the MIMIC-III database (Johnson et al., 2016). We simplify the real-world task by taking effective to mean that the organism is susceptible to the antibiotic. When treating patients for infections in the ICU, it is common that microbial cultures are tested for resistance. This presents a rare opportunity for off-policy policy evaluation, as the outcomes of these tests may be used as the ground truth potential outcomes of treatment (Boominathan et al., 2020). In practice, the results of these tests are not always available at the time of treatment. For this reason, we learn models based on the test outcomes *only of treatments actually given to patients*. To simplify further, we interpret concurrent treatments as sequential; their outcomes are not conflated here since they are taken from the culture tests. We stress that this task is not meant to accurately reflect clinical practice, but to serve as a benchmark based on a real-world distribution. Although a patient's condition may change as a response to treatment, bacteria typically do not develop resistance during a particular ICU stay, and so the stationarity assumption is valid.

Baseline covariates $X$ of a patient represent their age group (4 groups), whether they had infectious or skin diseases ($2 \times 2$ groups), and the identity of the organism, e.g., Staphylococcus aureus. These were found to be important predictors of resistance by Ghosh et al. (2019). In total, $X$ comprised 12 binary indicators. From the full set of microbial events in MIMIC-III, we restricted our study to a subset of 4 microorganisms and 6 antibiotics, selected based on overall prevalence and the rate of co-occurrence in the data. There were three distinct final outcomes of culture tests, *resistant, intermediate, susceptible*, encoded as $Y = 0, 1, 2$, respectively, where higher is better. The resulting cohort restricted to patients treated using only the selected antibiotics consisted of $n = 1362$ patients which had cultures tested for resistance against all antibiotics. The cohort was split randomly into a training and test set with a 70/30 ratio and experiments were repeated over five such splits. Patients treated for multiple organisms were split into different instances. A full list of variables, the selected antibiotics and organisms, and additional statistics are given in Appendix C.2.

We compare our learned policies to the policy used to select antibiotics in practice. However, due to censoring, e.g., from mortality, the sequence length of observed patients may not be representative of the expected number of trials used by the observed policy before an effective treatment is found. In other words, the average outcome for patients who went through $t$ treatments is a biased estimate of the value of the observed policy. Therefore, for direct comparison with current practice ("Doctor"), only the mean outcome following the first treatment point is displayed (star marker) in Figure 3a. For an approximate comparison with current practice, as used in multiple treatment trials, we created a

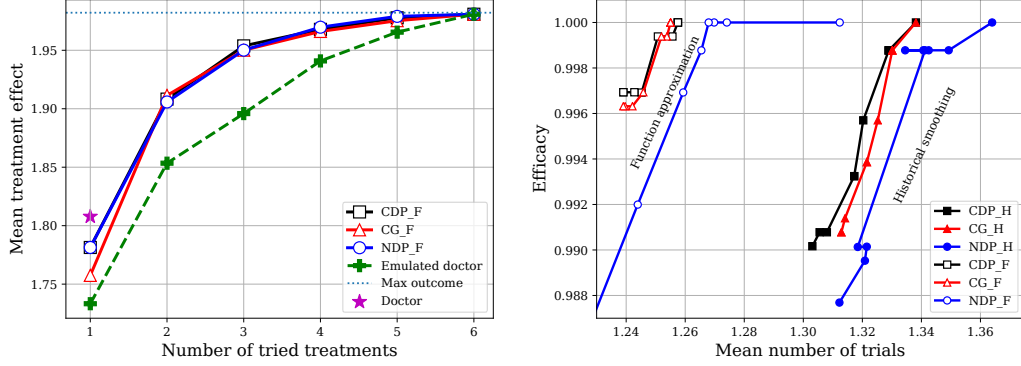

(a) Mean best-at-termination or best-so-far outcome found after a given number of trials, across all subjects, for different policies. $\delta = 0, \lambda = 0.35$.

(b) Efficacy of antibiotics vs the mean number of trials for different values of $\delta$ and $\lambda$ (one value per marker) and different model estimation schemes.

Figure 3: Results from the antibiotics experiment. Average best-found outcome of different policies across patients at different stages of the search (a) and efficacy and search time (number of treatment trials) as functions of $\delta$ (b). In plot (a), at a given number of trials, the best-so-far outcome is used for ongoing sequences, and the best-at-termination is used for terminated ones. Efficacy refers to the rate at which a near-optimal treatment is found at the given $\delta$. Suffixes _F and _H indicates model estimation using function approximation and historical smoothing respectively. $\epsilon = 0$.

baseline dubbed "Emulated doctor". It uses a tabular estimate of the observed policy to imitate the choices made by doctors in the dataset in terms of the history $H = (X, \overline{A}, \overline{Y})$, i.e., it operates on the same information as the other algorithms. We compare this to CDP, CG and NDP, and evaluate all policies using culture tests for held-out observations. We sweep all hyperparameters uniformly over 10 values; for CDP, CG, $\delta \in [0, 1]$, for NDP_H, $\lambda \in [0, 0.5]$ and for NDP_F, $\lambda \in [0, 1]$.

In Figure 3a, we see that CG, CDP and NDP, with function approximation, all learn comparable policies that are preferable to the estimated behavior policy. The mean search length was 1.26 for CDP and NDP, 1.28 for CG and 1.38 for Emulated doctor. We see that the best treatment found after a single trial is slightly better in the raw data (star marker). This may be because more information is available to the physician than to our algorithms. The physician could (1) take into account the original value of continuous variables, such as age, instead of using age groups and (2) use more features of the patient in order to find the right treatment. Using more covariates in this instance would make the problem impractical to solve without further approximations since the table generated by the dynamic programming algorithm grows exponentially. The current variable set was restricted for this reason. In Figure 3b, we see that across different values of $\delta, \lambda$, all algorithms achieve near-optimal efficacy (almost 1), but vary in their search time. CDP is equal or preferable to CG, with the model-free baseline NDP achieving the worst results. A much more noticeable difference is that between policies learned using the model estimated with function approximation (suffix _F) and those with a (smoothed) tabular representation (suffix _H).

# 7 Conclusion

We have formalized the problem of learning to search efficiently for causally effective treatments. We have given conditions under which the problem is solvable by learning from observational data, and proposed algorithms that estimate a causal model and perform policy optimization. Our solution using constrained dynamic programming problem (CDP) in an exponentially large state space illustrates the associated computational difficulties and prompted our investigation of two approximations, one based on greedy search and one on model-free reinforcement learning. We found that the greedy search algorithm performed comparably to the exact solution in experiments and was less sensitive to sample size. Determining conditions under which greedy algorithms are preferable statistically is an interesting open question. We believe that our work will have the largest impact in settings where a) the assumption of potential outcome stationarity is justified, b) even a small reduction in search time is valuable and c) a transparent trade-off between efficacy and search time is valuable in itself.

## Broader impact

Personalized and partially automated selection of medical treatments is a long-standing goal for machine learning and statistics with the potential to improve the lives of patients and reduce the workload on physicians. This task is not without risk however, as poor decisions may fail to reduce or even increase suffering. It is important that implementations of such ideas is guided by strong domain knowledge, thorough evaluation and that checks and balances are in place. Many previous works in this field aim to identify new policies for treatment or doses with the goal of improving treatment response itself. This goal is not always feasible to achieve—some conditions are fundamentally hard to treat with available medications and procedures. In contrast, we focus on conditions where a good enough treatment would be identified by an existing policy given enough time, with the goal of reducing this search time as much as possible. The trade-off between a good outcome and time is made transparent using a model of patient outcomes and a certainty parameter. With this, we hope to contribute towards making machine learning methods more suitable for clinical implementation.

## Funding disclosure

This work was supported in part by the Wallenberg AI, Autonomous Systems and Software Program (WASP) funded by the Knut and Alice Wallenberg Foundation.

## Footnotes

[3]Implementations can be found at: `https://github.com/Healthy-AI/TreatmentExploration`

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
