[Supplementary Material]

# Supplementary material for: Learning to search efficiently for causally near-optimal treatments

**Samuel Håkansson**
University of Gothenburg
samuel.hakansson@gu.se

**Viktor Lindblom**
Chalmers University of Technology
viklindb@student.chalmers.se

**Omer Gottesman**
Brown University
omer_gottesman@brown.edu

**Fredrik D. Johansson**
Chalmers University of Technology
fredrik.johansson@chalmers.se

## A   Proofs of theorems

### A.1   Proof of Lemma 1 (Stationarity)

**Lemma S1** (Lemma 1 restated). *Let $\mathcal{I}$ be a permutation of the sequence $(1, ..., s)$. Then, for our causal graph under Assumption 2, for $b \in \mathcal{A}$,*

$$p(Y(b) \mid X, \overline{A}_s = \overline{a}, \overline{Y}_s = \overline{y}) = p(Y(b) \mid X, \overline{A}_s = (a_{\mathcal{I}(1)}, ..., a_{\mathcal{I}(s)}), \overline{Y}_s = (y_{\mathcal{I}(1)}, ..., y_{\mathcal{I}(s)}))$$

*Proof.* Let $h = (x, (a_1, y_1), ..., (a_s, y_s))$. Let $\pi$ be a permutation of $1, ..., s$ and $\pi(r)$ the index assigned to $r$. We use the short-hands $p(a) = p(A = a)$, $p(A \mid b) = p(A \mid B = b)$, etc.

$$p(Y(a) \mid H_s = h_s, A_s = a_s) \qquad \text{stationarity}$$

$$= \frac{p(Y_s(a), h_s, a_s)}{p(h_s, a_s)}$$

$$= \frac{\sum_z p(Y_s(a), h_s, a_s, z)}{\sum_z p(h_s, a_s, z)} \qquad \text{prob. laws}$$

$$= \frac{\sum_z p(Y_s(a) \mid h_s, a_s, z) p(a_s \mid h_s, z) p(h_s \mid z) p(z)}{\sum_z p(a_s \mid h_s, z) p(h_s \mid z) p(z)} \qquad \text{expand}$$

$$= \frac{\sum_z p(Y_s(a) \mid h_s, a_s, z) p(a_s \mid h_s, z) \prod_r p(y_r \mid h_r, a_r, z) p(a_r \mid h_r, z) p(z)}{\sum_z \prod_r p(y_r \mid h_r, a_r, z) p(a_r \mid h_r, z) p(z)} \qquad \text{expand history}$$

$$= \frac{\sum_z p(Y_s(a) \mid a_s, z) p(a_s \mid h_s) \prod_r p(y_r \mid a_r, z) p(a_r \mid h_r) p(z)}{\sum_z p(a_s \mid h_s) \prod_r p(y_r \mid a_r, z) p(a_r \mid h_r) p(z)} \qquad A_s \perp\!\!\!\perp Z \mid H_s$$

$$= \frac{\sum_z p(Y_s(a) \mid z) \prod_r p(y_r(a_r) \mid z) p(z)}{\sum_z \prod_r p(y_r(a_r) \mid z) p(z)} \qquad \text{cancel terms}$$

$$= \frac{\sum_z p(Y(a) \mid z) \prod_r p(Y_{\pi(r)}(a_r) = y_r \mid z) p(z)}{\sum_z \prod_r p(Y_{\pi(r)}(a_r) = y_r \mid z) p(z)} \qquad \text{stationarity}$$

$$\square$$

Since the last expression is invariant to $\pi$, the result follows.

### A.2   Proof of Theorem 1 (Identifiability)

**Theorem S1** (Theorem 1 restated). *Under Assumptions 1–4, the stopping statistic $\rho(h)$ in (2) and $\epsilon, \delta$-optimality are identifiable from the observational distribution $p(X, T, \overline{A}, \overline{Y})$. In particular, for*

*any time step $s$ with history $h_s$, let $h(\mathcal{I})_s = (x, a_{\mathcal{I}(1)}, ..., a_{\mathcal{I}(s)}, y_{\mathcal{I}(1)}, ..., y_{\mathcal{I}(s)})$ be an arbitrary permutation of $h_s$. Then, for any sequence of untried (future) actions $\overline{a}_{s+1:k} = (a_{s+1}, ..., a_k) \in \mathcal{S}(\mathcal{A}_{-h_s})$ with $h(\mathcal{I})_r$ the continued history at time $r > s$ corresponding to $\overline{a}_{s+1:k}$ and $\overline{y}_{s+1:k}$,*

$$\rho(h_s) = \sum_{\overline{y}_{s+1:k} \in \mathcal{Y}^{k-s}} \mathbb{1}\left[\max(\overline{y}) > \mu(h_s) + \epsilon\right] \prod_{r=s+1}^{k} p(Y_r = y_r \mid A_r = a_r, H_r = h(\mathcal{I})_{r-1}) . \quad \text{(S1)}$$

*where $\mu(h_s) = \max_{(a,y) \in h_s} y$.*

*Proof.* Fix any history $h = (x, (a_1, y_1), ..., (a_s, y_s)) \in \mathcal{H}$ with $s = |h|$, any time points $q, r \in [k]$, any $a \in \mathcal{A}$ and let $\overline{a} \in \mathcal{S}(\mathcal{A})$ such that the subsequence $\overline{a}_s = (a_1, ..., a_s)$ coincides with $h$. Then, by Assumption 2, we have

$$Y_r(a) = Y_q(a) = Y(a) \quad \text{and} \quad \max_{a \notin h} Y(a) = \max_{r=s+1}^{k} Y_r(a_r) .$$

Below, we sum over sequences of outcomes $\overline{y}_{s+1:k} = (y_{s+1}, ..., y_k) \in \mathcal{Y}^{k-s}$ and refer to the history $h_r$ for $r > s$. Here, $h_r = (x, (a_1, y_1), ..., (a_r, y_r))$ is a sequence of both observed actions and outcomes (corresponding to the sub-sequence $h_s \subseteq h_r$) and unobserved ones. By definition, we have for any sequence of actions $\overline{a} \in S(\mathcal{A})$ according to the above, for any $\mu \in \mathcal{Y}$

$$\rho_\mu(h_s) = \sum_{\overline{y}_{s+1:k} \in \mathcal{Y}^{k-s}} p([Y(a_{s+1}), \ldots, Y(a_k)] = \overline{y}_{s+1:k} \mid H_s = h_s) \mathbb{1}[\max(\overline{y}_{s+1:k}) > \mu]$$

$$= \sum_{\overline{y}_{s+1:k} \in \mathcal{Y}^{k-s}} \mathbb{1}\left[\max(\overline{y}_{s+1:k}) > \mu\right] \prod_{r=s+1}^{k} p(Y_r(a_r) = y_r \mid H_{r-1} = h_{r-1})$$

$$= \sum_{\overline{y}_{s+1:k} \in \mathcal{Y}^{k-s}} \mathbb{1}\left[\max(\overline{y}_{s+1:k}) > \mu\right] \prod_{r=s+1}^{k} p(Y_r = y_r \mid A_r = a_r, H_{r-1} = h_{r-1}) .$$

In the second step we apply Assumption 2 (stationarity) and in the third Assumptions 1–Assumptions 4 (consistency, sequential ignorability). Finally, from ignorability and stationarity, we have for any permutations $h(\mathcal{I})_s$,

$$\rho_\mu(h_s) = \sum_{\overline{y}_{s+1:k} \in \mathcal{Y}^{k-s}} \mathbb{1}\left[\max(\overline{y}_{s+1:k}) > \mu\right] \prod_{r=s+1}^{k} p(Y_r = y_r \mid A_r = a_r, H_{r-1} = h(\mathcal{I})_{r-1}) .$$

Doing so, we obtain the result in (4). In solving (1), we only need to evaluate $\rho(h_s)$ for histories with positive support under $p_\pi$. Assumption 3 (positivity) ensures that there exists at least one permutation $\overline{a} \in \mathcal{S}(\mathcal{A}_{-h_s})$ such that $p(A_{s+1:k} = \overline{a} \mid H_s = h_s)$. This in turn implies identifiability. $\square$

### A.3 Bounds on stopping criterion

**Theorem S2.** *For any threshold $\mu \in \mathcal{Y}$ and history $h \in \mathcal{H}$, we have under Assumption 2,*

$$\underbrace{\max_{a \notin h}\left[p\left(Y(a) > \mu \mid h\right)\right]}_{\text{Used for less conservative stopping}} \leq \underbrace{p\left(\max_{a \notin h} Y(a) > \mu \mid h\right)}_{=:\, \rho_\mu(h)} \leq \underbrace{\sum_{a \notin h} p\left(Y(a) > \mu \mid h\right)}_{\text{Used for more conservative stopping}} \quad \text{(S2)}$$

*Proof.* Let $\mathcal{A}_{-h} = \{a \in \mathcal{A} : a \notin h\}$. We start with the upper bound. By definition

$$\{\overline{y} \in \mathcal{Y}^{|\mathcal{A}_{-h}|} : \max(\overline{y}) > \mu\} = \bigcup_{a \in \mathcal{A}_{-h}} \{\overline{y} \in \mathcal{Y}^{|\mathcal{A}_{-h}|} : y_a > \mu\}$$

Hence, by Boole's inequality,

$$p\left(\max_{a \notin h} Y(a) > \mu \mid H = h\right) \leq \sum_{a \in \mathcal{A}_{-h}} \sum_{\overline{y} \in \mathcal{Y}^{|\mathcal{A}_{-h}|} :\, y(a) > \mu} p\left(Y(\mathcal{A}_{-h}) = \overline{y} \mid H = h\right)$$

$$= \sum_{a \in \mathcal{A}_{-h}} p\left(Y(a) > \mu \mid h\right) .$$

For the lower bound, the argument is equally straight-forward.

$$p\left(\max_{a\notin h} Y(a) > \mu \mid H = h\right) = \sum_{\overline{y}:\ \max(\overline{y})>\mu} p\left(Y(\mathcal{A}_{-h}) = \overline{y} \mid H = h\right)$$

$$\geq \max_{a\in\mathcal{A}_{-h}} \sum_{\overline{y}:\ y(a)>\mu} p\left(Y(\mathcal{A}_{-h}) = \overline{y} \mid H = h\right)$$

$$= \max_{a\in\mathcal{A}_{-h}} p\left(Y(a) > \mu \mid H = h\right).$$

$\square$

## A.4 Proof of Theorem 2

We restate the following assumption and Theorem 2 for convenience.

**Assumption S1.** *A random variable $U$ has $\alpha$-bounded propensity sensitivity relative to $H$ if for all $u \in \mathcal{U}, h \in \mathcal{H}$, with $s = |h|$ and $\overline{a} \in \mathcal{A}^{k-s}$, for some $\alpha \geq 1$, with $\overline{A}_{s+1:k} = (A_{s+1}, ..., A_k)$,*

$$\frac{1}{\alpha} \leq \frac{\Pr[\overline{A}_{s+1:k} = \overline{a} \mid H = h]}{\Pr[\overline{A}_{s+1:k} = \overline{a} \mid U = u, H = h]} \leq \alpha .$$

**Theorem S3** (Theorem 2 restated). *Given is that Assumption S1 (bounded propensity) holds for $H, U$ with sensitivity parameter $\alpha \geq 1$ and Assumption 4 (ignorability) holds for all $s \in [k]$ w.r.t. confounders $(H_s, U)$. Let $Y_r$ be the (hypothetical) outcome of treatment $A_r$ at time $r = s + 1, ..., k$. Then, for any history $h \in \mathcal{H}_s$ and the set of treatments $\overline{a} = \mathcal{A} \setminus \mathcal{A}(h)$, it holds that*

$$\Pr[\max_{r=s+1}^{k} Y_r > \mu \mid \overline{A}_{s+1:k} = \overline{a}, H_s = h] \leq \frac{\delta}{\alpha} \implies \Pr[\max_{a\in\overline{a}} Y(a) > \mu \mid H_s = h] \leq \delta .$$

*Proof.* We have by definition, where $\overline{y} > \mu$ applies element-wise,

$$\Pr[\max_{a\in\overline{a}} Y(a) > \mu \mid H = h] = \sum_{\overline{y}:\overline{y}>\mu} \Pr[\overline{Y}(\overline{a}) = \overline{y} \mid H = h]$$

$$\Pr[\max_{i} Y_i > \mu \mid \overline{A} = \overline{a}, H = h] = \sum_{\overline{y}:\overline{y}>\mu} \Pr[\overline{Y} = \overline{y} \mid \overline{A} = \overline{a}, H = h]$$

Then, marginalizing over the unobserved confounder $U$ and conditioning on $H$,

$$\Pr[\max_{a\in\overline{a}} Y(a) > \mu \mid H = h] = \sum_{\substack{\overline{y}:\overline{y}>\mu \\ u\in\mathcal{U}}} \Pr[\overline{Y}(\overline{a}) = \overline{y} \mid h, U = u] p(U = u \mid h)$$

$$= \sum_{\substack{\overline{y}:\overline{y}>\mu \\ u\in\mathcal{U}}} \Pr[\overline{Y} = \overline{y} \mid h, u, \overline{a}] p(u \mid h)$$

where the last equality follows from ignorability w.r.t. $H, U$. Applying the same steps to $\Pr[\max_i Y_i > \mu \mid \overline{A} = \overline{a}]$, we get

$$\Pr[\max_{i} Y_i > \mu \mid h, \overline{A} = \overline{a}] = \sum_{\substack{\overline{y}:\overline{y}>\mu \\ u\in\mathcal{U}}} \Pr[\overline{Y} = \overline{y} \mid h, u, \overline{a}] p(u \mid \overline{a}, h)$$

We find that

$$\Pr[\max_{a\in\overline{a}} Y(a) > \mu \mid h] - \Pr[\max_{i} Y_i > \mu \mid h, \overline{A} = \overline{a}]$$

$$= \sum_{\substack{\overline{y}:\overline{y}>\mu \\ u\in\mathcal{U}}} \Pr[\overline{y} \mid h, u, \overline{a}] \left(p(u \mid h) - p(u \mid \overline{a}, h)\right)$$

$$= \sum_{\substack{\overline{y}:\overline{y}>\mu \\ u\in\mathcal{U}}} \Pr[\overline{y} \mid h, u, \overline{a}] p(u \mid \overline{a}, h) \left(\frac{p(u \mid h)}{p(u \mid \overline{a}, h)} - 1\right) = (*)$$

By Bayes rule, we have

$$\frac{p(u \mid h)}{p(u \mid \overline{a}, h)} = \frac{p(u \mid h)p(\overline{a} \mid h)}{p(\overline{a} \mid u, h)p(u \mid h)} = \frac{p(\overline{a} \mid h)}{p(\overline{a} \mid u, h)}$$

and so,

$$(*) = \sum_{\substack{\overline{y}:\overline{y}>\mu \\ u \in \mathcal{U}}} \Pr[\overline{y} \mid h, u, \overline{a}]p(u \mid \overline{a}, h) \left( \frac{p(\overline{a} \mid h)}{p(\overline{a} \mid u, h)} - 1 \right)$$

The result follows immediately from our Assumption S1, that $\frac{1}{\alpha} \leq \frac{p(\overline{a}|h)}{p(\overline{a}|u,h)} \leq \alpha$. In fact, only the upper bound is needed. $\square$

## A.5   Proof of Theorem 3 (Correctness of dynamic programming)

**Theorem S4** (Theorem 3 restated). *Recall that $H_0 = (X, \emptyset, \emptyset)$. The policy maximizing* (6), $\pi(h) = \arg\max_a Q(h, a)$, *is an optimal solution to* (1) *and its expected search time is* $\mathbb{E}[T] = \mathbb{E}_X[-V(H_0)]$.

*Proof sketch.* Recall that

$$\gamma_{\epsilon,\delta,\alpha}(h) := \mathbb{1}[\Pr[\max_{a' \notin h} Y(a') > \mu(h) + \epsilon \mid H = h] < \delta/\alpha] \tag{S3}$$

$$Q(h, a) = r(h, a) + \mathbb{1}[a \neq \text{STOP}] \sum_{y \in \mathcal{Y}} p(Y(a) = y \mid h) \max_{a' \in \mathcal{A} \cup \{\text{STOP}\}} Q(h \cup \{(a, y)\}, a') , \tag{S4}$$

$$r_{\epsilon,\delta,\alpha}(h, a) = \begin{cases} -\infty, & \text{if } a = \text{STOP}, \gamma_{\epsilon,\delta,\alpha}(h) = 0 \\ 0, & \text{if } a = \text{STOP}, \gamma_{\epsilon,\delta,\alpha}(h) = 1 \\ -1, & \text{if } a \neq \text{STOP} \end{cases} . \tag{S5}$$

and $V(h) = \max_{a,c} Q(h, a)$.

By definition, any policy that achieves a finite expected reward $\mathbb{E}_{H_0}[V(H_0)]$ satisfies the stopping criterion, and is therefore a *feasible solution* to (1). Furthermore, any time search is terminated ($a = \text{STOP}$ or $\mathcal{A}_{-h} = \emptyset$), the expected sum of rewards for a sequence is equal to minus the number of steps spent until the sequence terminates. The sequence is optimal if it terminates as soon as an $\epsilon, \delta$-optimal treatment is found. Thus, a policy with finite expected return that maximizes $V(H_0) = \max_a Q(H_0, a)$ is an optimally efficient search policy for effective treatments. $\square$

## A.6   Approximation ratio of greedy algorithms

The active learning problem concerns identification of a hypothesis $g \in \mathcal{G}$ by iteratively performing tests suggested by a policy (Guillory and Bilmes, 2009). The problem then amounts to finding a policy $\pi$ which selects tests $\overline{A} = A_1, ..., A_T$, the results $Y(A_1), ..., Y(A_T)$ of which identify $g$ with probability 1, $p(G = g \mid Y(A_1), ..., Y(A_T)) = 1$. We consider now the case were a prior distribution $p(G, Y(1), ..., Y(k))$ is known, as studied by (Guillory and Bilmes, 2009). A sequence of tests $\overline{A}$ which identifies $g$ is associated with a cost $c(\overline{A}, G)$, and the objective is to find $\pi$ which minimizes the expected cost over $p$,

$$c(\pi) = \mathbb{E}_{G, A \sim \pi}[c(\overline{A}, G)] .$$

We have the following result from the literature.

**Theorem S5** (Adapted from Theorem 4 of (Kosaraju et al., 1999)). *There exists a greedy policy $\pi$ such that for any $p$ such that $Y(a) : a \in \mathcal{A}$ are deterministic given $G$,*

$$c(\pi) \leq c(\pi^*)O(\log|\mathcal{G}|)$$

*where $\pi^* = \arg\min_{\pi'} c(\pi')$.*

This bound is matched by a lower bound by (Chakaravarthy et al., 2007) which states that it is NP-hard to achieve an approximation ratio better than $o(\log |\mathcal{G}|)$.

In the setting with $\delta = 0$, our problem may posed as active learning where the hypothesis corresponds to the maximum value of potential outcomes, $G = \min\{g \in \mathcal{Y} \colon p(\max_a Y(a) > g) \le 0\}$. Once this quantity is identified, the stopping criterion may be determined immediately. However, under this hypothesis, $Y(a)$ are not deterministic given $G$ and the results above do not apply. Golovin et al. (2010) study the noisy case under the assumption that non-determinism in $Y(a)$ is controlled by a noise variable $\Theta$, i.e., that $\overline{Y}(\mathcal{A}) = f(G, \Theta)$ for some deterministic function $f$.

**Theorem S6** (Adapted from Theorem 3 in (Golovin et al., 2010) with uniform costs)**.** *Fix hypotheses $\mathcal{G}$, tests $\mathcal{A}$ and outcomes in $\mathcal{Y}$, Fix a prior $p(G, \Theta)$ and a function $f \colon G \times \mathrm{supp}(\Theta) \to \mathcal{Y}^{|\mathcal{A}|}$ which define the probabilistic noise model. Let $c(\pi)$ denote the expected cost of $\pi$ incurs to identify which equivalence class $G$ the outcome vector $\overline{Y}(A_T)$ belongs to. Let $\pi^*$ denote the policy minimizing $c(\cdot)$, and let $\pi$ denote the adaptive policy implemented by the greedy algorithm EC2. Then,*

$$c(\pi) \le c(\pi^*) O(\log |\mathcal{A}| + \log |\mathrm{supp}(\Theta)|) \,.$$

In the case that all combinations of outcomes are feasible, $\log |\mathrm{supp}(\Theta)| = |\mathcal{A}| \log |\mathcal{Y}|$ and the bound above is vacuous, since a trivial bound on the search time is $|\mathcal{A}|$. When there is structure in potential outcomes, $\mathrm{supp}(\Theta)$ may be much smaller. For example, if the moderating variable $Z$ controls all uncertainty in $Y(a)$, given X, the bound reduces to $O(\log |\mathcal{Z}_X|)$ where $\mathcal{Z}_X = \{z \in \mathcal{Z} \colon p(Z \mid X) > 0\}$, which may be significantly smaller than $|\mathcal{A}| \log |\mathcal{Y}|$.

## A.7 Model-free RL and CDP are not equivalent

Let $\max_{(\cdot, y) \in h} y$ represent the best outcome so far at history $h$, with $s = |h|$ and $\lambda > 0$ a parameter trading off early termination and high outcome. Now, consider the reward function $r_\lambda^{\text{model-free}}(h, a)$ following history $h \in \mathcal{H}$ defined below.

$$r_\lambda^{\text{model-free}}(h, a) = \begin{cases} 0, & a \ne \textbf{STOP} \\ \max_{(\cdot, y) \in h} y - \lambda|h|, & a = \textbf{STOP}. \end{cases} \quad \text{(S6)}$$

and the policy maximizing the expected sum of rewards

$$\pi_\lambda^{*,\text{model-free}} = \arg\max_\pi \mathbb{E}_{h, a \sim \pi}\left[\sum_{s=1}^k r_\lambda^{\text{model-free}}(h_s, a_s)\right] \,. \quad \text{(S7)}$$

Now consider the greedy policy maximizing the Q-function defined by

$$Q(h, a) = \mathbb{E}_{h' | h, a}[r_\lambda^{\text{model-free}}(h, a) + \max_{a' \in \mathcal{A}_{-h} \cup \{\textbf{STOP}\}} Q(h', a') \mid H_s = h, A_s = a] \,. \quad \text{(S8)}$$

For readers familiar with reinforcement learning, it is easy to see that policy maximizing $Q$ defined above also maximizes the expected sum of rewards given by (S6). Below, we prove that this algorithm does not in general solve (1).

**Theorem S7.** *There are instances of (1) (main problem), specified by a distribution $p$ and parameters $\epsilon, \delta$, such that the solutions to (1) and (S7) are distinct for every choice of $\lambda > 0$.*

*Proof.* Consider a context-less setting with two actions $\mathcal{A} = \{a, b\}$ with the following potential outcomes: $p(Y(a) = 1.0) = 1/2, p(Y(a) = 0.5) = 1/2$ and $p(Y(b) = 0.5 + \epsilon) = 1$. In this scenario, having observed nothing, the probability that action $b$ yields a higher outcome than $a$ is 1/2. Hence, for $\delta = 0.5$, CDP always prefers to start with action $b$ and end immediately. Now, consider NDL, which minimizes the expected return with the reward function,

$$r(h, a) = \begin{cases} 0, & a \ne \textbf{STOP} \\ \max_{(\cdot, y) \in h} y - \lambda|h|, & a = \textbf{STOP} \end{cases}$$

where $s$ indicates the stop action and $\max_{(\cdot, y) \in h} y$ represents the best outcome so far at history $h$ and $\lambda > 0$. The Q-function is in (S8). NDP computes this recursively and uses the policy which maximizes it. Under the version of this problem with $\epsilon < 0.25$, we can show that there is no $\lambda > 0$ such that $Q(\emptyset, b) > Q(\emptyset, a)$. We give the map of $Q$ below under this assumption.

For $\lambda > \epsilon$, $Q(\emptyset, a) = 0.75 - \lambda$ and $Q(\emptyset, b) = \max(0.5 + \epsilon - \lambda, 0.75 + \epsilon/2 - 2\lambda) < Q(\emptyset, a)$. For $0 < \lambda \le \epsilon$, we have $Q(\emptyset, a) = 0.75 - 1.5\lambda + \epsilon/2 > Q(\emptyset, b)$ by the assumption $\epsilon < 0.25$. Hence, NDL would, for any $\lambda$ prefer action $a$. However, for $\delta = 0.5$, CDP would prefer action $b$. Thus, for $\delta = 0.5$, there is no $\lambda$ which make these equivalent. $\qquad \square$

| $A_1$ | $Y_1$ | $A_2$ | $Y_2$ | $a$ | $Q(h,a)$ |
|---|---|---|---|---|---|
| a | 1.0 | – | – | **STOP** | $1.0 - \lambda$ |
| a | 0.5 | – | – | **STOP** | $0.5 - \lambda$ |
| b | $0.5 + \epsilon$ | – | – | **STOP** | $0.5 + \epsilon - \lambda$ |
| a | 1.0 | b | $0.5 + \epsilon$ | **STOP** | $1.0 - 2\lambda$ |
| a | 0.5 | b | $0.5 + \epsilon$ | **STOP** | $0.5 + \epsilon - 2\lambda$ |
| a | 1.0 | – | – | b | $1.0 - 2\lambda$ |
| a | 0.5 | – | – | b | $0.5 + \epsilon - 2\lambda$ |
| b | $0.5 + \epsilon$ | – | – | a | $\frac{(1.0-2\lambda)+(0.5+\epsilon-2\lambda)}{2}$ |
| – | – | – | – | a | $\frac{(1.0-\lambda)+\max(0.5-\lambda,0.5+\epsilon-2\lambda)}{2}$ |
| – | – | – | – | b | $\max(0.5 + \epsilon - \lambda, \frac{(1.0-2\lambda)+(0.5+\epsilon-2\lambda)}{2})$ |

*(the table header shows "$h$" spanning $A_1, Y_1, A_2, Y_2$)*

## B  Historical smoothing and function approximation

The number of possible combinations (histories) grows exponentially with the number of actions, $k = |\mathcal{A}|$. As a result, it is very probably that certain combinations of histories $h$ and actions $a$ are never observed in practice. We consider two solutions to this: historical smoothing and function approximation. Historical smoothing is used in the discrete case by estimating the probability $p(Y(a) = y \mid H_{s-1} = h)$ using a weighted average of outcomes for observations $(h, a, y)$ and observations for subsequences $(h', a, y)$ where $h' \subseteq h$. Function approximation imputes $\hat{p}(Y(a) = y \mid H_{s-1} = h)$ using a regression estimator trained on all observations. We expand on these approaches in Appendix B.

### B.1  Historical smoothing

Consider estimating the function $p(Y(a) \mid H = h)$ in the discrete case. Under the stationarity assumption, Assumption 2, it is sufficient to represent the history in terms of indicators for tried treatments, $\{B_a \colon a \in \mathcal{A}\}$ such that $B_a \in \{0, 1\}$, and observed outcomes of these actions. Hence, $p(Y(a) \mid H = h)$ may be represented by a table of dimensions $|\mathcal{Y}| \times (\{0, 1\} \times |\mathcal{Y}|)^{|\mathcal{A}|}$. Clearly, even under this representation, the number of possible histories grows exponentially with the number of actions. For this reason, for moderate to high numbers of actions, it will be unlikely to observe samples for each cell of this table.

To obtain an estimate even in cases with high dimensionality, we use historical smoothing based on a *prior*. In the discrete case, we may view the distribution of the outcomes $Y(a)$ for a treatment $a$ following history $h$ as a categorical distribution. We impose a Dirichlet prior on this distribution and use the posterior distribution in estimating the stopping statistic $\rho$ and in policy optimization. A Dirichlet prior for $p(Y(a) \mid H = h)$ is specified by pseudo-counts $\beta_1(a, h), ..., \beta_{|\mathcal{Y}|}(a, h)$. The posterior parameters are then $\frac{n_y(a,h)+\beta_y(a,h)}{\sum_{y'} n_{y'}(a,h)+\beta_{y'}(a,h)}$, where $n_y(a, h)$ is equal to the number of samples where $Y(a) = y$ following history $h$. In this work, we consider two different priors $\beta$.

**Historical prior (kernel smoothing)**   The historical prior assumes that the conditional outcome distribution changes slowly with the number of past observations. The prior itself is a weighted average of the outcome probability at all possible previous histories,

$$\beta_y(a, h) = \sum_{h' \subset h} w(h, h') \cdot \hat{p}(Y(a) \mid H = h'), \qquad (S9)$$

where the weight of the probability given by a shorter history is determined by its similarity to $h$,

$$w(h, h') = \frac{e^{-(|h|-|h'|-1)^2}}{|h| \cdot 2^{|h-h_i|-1}} . \qquad (S10)$$

**Uninformed prior**   The uninformed prior assigns a small uniform value to all $\beta$.

## B.2 Function approximation

Observations for the $i$th subject are denoted $x^{(i)}, a_t^{(i)}, y_t^{(i)}, \bar{a}_s^{(i)}$. To use function approximation, we fit a single function $f$, acting on a representation of history $\phi(h)$ to estimate $p(Y(a) \mid H = h)$ by solving the following problem,

$$\min_{f \in \mathcal{F}} \sum_{i=1}^{n} \sum_{s=1}^{t_i} L(f(h_s^{(i)}, a_s^{(i)}), y_s^{(i)}) , \tag{S11}$$

for an appropriately chosen function class $\mathcal{F}$ and loss function $L$. In the discrete settings considered in the paper, we use the logistic (cross-entropy) loss which leaves the solution to (S11) a probabilistic classifier, or estimate of $p(Y(a) \mid H = h)$ for all $a, h$.

# C  Additional experimental results

Below follow additional details and results from the experiments. All experiments were implemented in Python and run on standard laptop computers. Each experiment on the synthetic DGP took less than a handful of hours to finish. For the antibiotics experiment, the overall time to produce the results for all values of $\delta$ was 2 days.

## C.1  Synthetic data generating process

We describe the datagenerating process (DGP) for the synthetic dataset used in Figure 2b and additional results described below. Let $o_a(h) = \mathbb{1}[a \in h]$ and $o(h) = [o_1(h), ..., o_k(h)]^\top$. The moderator $Z \in \{0, 1\}^d$ and covariates $X \in \{0, 1\}^v$ are drawn according to

1. $Z \sim \text{Bernoulli}(\alpha)$
2. $X \sim \text{Bernoulli}(\max(\min(\beta Z, 0.98), 0.02))$.

given a set of parameters $\alpha \in [0, 1]^d, \beta \in [0, 1]^{v \times d}$ drawn element-wise uniformly at random.

The action **STOP** is drawn at any point following the first treatment with probability $p_{\textbf{STOP}} = 0.1$. To emulate a closer-to-realistic policy, if not stopped, the next action is drawn according to a categorical distribution with probabilities determined by the variable $X$ and the dissimilarity of the new action $A$ to previous actions in $H$. Outcomes are drawn according to a categorical distribution with parameters given by the pdf of a Cauchy random variable, itself with parameters depending on the variables $X$, $Z$ and $A$. For a full description of the data generating distribution, see Algorithm 1.

### C.1.1  Additional results for the synthetic DGP

We present additional results for CDP, CG and NDP applied to the synthetic DGP described above. Unless otherwise specified, $\delta = 0.4, \epsilon = 0, \lambda = 0.35$ and CDP and CG use the upper bound approximation of the stopping criterion described in Appendix A.3 with historical smoothing (_H), as described in Appendix B.

In Figure S1, we illustrate the mean efficacy and search time (number of trials) as a function dataset size, varying logarithmically from $n = 50$ to $n = 75000$ samples. We include the variance across $m$ random seeds for the experiment, $\hat{\sigma}^2 = \frac{1}{m-1} \sum_{i=1}^{m} (x_i - \bar{x})^2$. This Figure is a different view of Figure 2b, where we clearly see that the efficacy for most algorithms go up as data set size grows and search time decreases. For NDP, as noted in Section 6, we see the opposite trend, however.

Figure S2 shows the trade-off between search time (number of trials) for different algorithms and 40 different values of $\delta \in [0, 1]$ with $\lambda = \delta$ for $n = 15000$ samples, in the setting corresponding to Figure 2b. In Figure S3, we give the corresponding comparison for using lower or upper bounds in the estimation of the stopping criterion $\rho$, as described in Appendix A.3. Here, _U refers to the upper bound, _L to the lower bound and _E is "exact" estimator, i.e. the empirical estimator of the exact expression for the stopping criterion, $\rho$. At first glance, the output of the different algorithms using different bounds appear very similar. However, as we see in Figure S4, the trade-off induced by a specific value of $\delta$ varies greatly depending on the estimation strategy. This is discussed also in Section 6, where we note that the policy learned by NDP is very sensitive to the setting of $\lambda$.

Figure S1: Efficacy and time over different sized training sets for the synthetic DGP. Interval widths represent the variance across 50 realizations.

Figure S2: Efficacy and search time (number of trials) for different policy optimization methods operating the same model (historical smoothing, upper bound).

(a) Constrained Dynamic Programming algorithm.          (b) Constrained Greedy algorithm.

Figure S3: Results using estimates of the stopping criterion based on the upper (_U) and lower bounds (_L) described in Appendix A.3, as well as the no-bound (exact) estimate (_E) for the CDP and CG algorithms with $\delta$ varying linearly in $[0, 1]$.

(a) Efficacy and search time (number of trials) for varying approximations used in estimating the stopping criterion, with the upper bound, in the CDP algorithm. _U stands for using a uniform prior to fill in missing valus. _H is the historical kernel smoothing described in Appendix B. _F refers to function approximation and _T the result for using the true model.

(b) Efficacy and search time (number of trials) when using different bounds on the stopping criterion $\rho$ in the CDP algorithm. _U stands for using the upper bound, _L for the lower bound and _E for the exact (no bound) estimate of $\rho$.

Figure S4: Efficacy and mean search time (number of trials), varying $\delta$ in $[0, 1]$.

## C.2  Antibiotic resistance dataset

Below, we give additional information on the antibiotic resistance dataset compiled from MIMIC-III.

To gather a cohort for which a consistent set of culture tests had all been performed for every patient, the set of organisms were restricted to a small subset. This selection was made based on overall prevalence in the data as well as the co-occurrence with common antibiotic culture tests. The selected organisms and antibiotics are listed below.

**Selected (bacterial) microorganisms:**

- Escherichia Coli (E. coli)
- Pseudomonas aeruginosa
- Klebsiella pneumoniae
- Proteus mirabilis

**Selected antibiotics:**

- Ceftazidime
- Piperacillin/Tazo
- Cefepime

(a)            (b)

Figure S5: Efficacy and mean search time over different values of $\delta$ on the antibiotic resistance data set. The width of the plots represent the unbiased empirical sample variance across random splits.

(a)            (b)

Figure S6: Efficacy and mean number of trials over different values of $\lambda$ for the Naive Dynamic Programming algorithm. Variance is unbiased sample variance across random splits of the data. $\lambda$ is perturbed by 0.0001 in order to avoid division by zero for $\lambda = 0$.

- Tobramycin
- Gentamicin
- Meropenem

*pending* was also an "result" in MIMIC-III, there were few of these instances and they were removed. Covariates X: Ages are divided into the four groups $[0, 15]$, $(15, 31]$, $(31, 60]$, and $(60, \infty)$. The two diseases are *Infectious And Parasitic Diseases* and *Diseases Of The Skin And Subcutaneous Tissue* as classified by ICD (WHO, 1978). The data was split in training and test 70/30 from 1362 patients and patients with multiple organisms were not split between the sets. Patients who had taken any antibiotic other than our chosen ones were not included in the data. Figure S5 uses the same data as Figure 3b but is split by $\delta$ and variance is shown.

| # of treatments | # of patients |
|:---:|:---:|
| 1 | 860 |
| 2 | 340 |
| 3 | 137 |
| 4 | 22 |
| 5 | 3 |

---

**Algorithm 1:** Generating distribution of actions and potential outcomes

---

**Input:** Weight parameter $w_x$ (default value 1)
**Input:** Number of outcomes $n_y$
**Input:** Uniform stopping probability $p_{\textbf{STOP}}$

Generating parameters:
$u_1, u_2 \sim \mathcal{N}(0_{k \times (1+v+d)}, 1)$
$u_2 \leftarrow |u_2|$
**for** $i \leftarrow 2$ **to** $v+1$ **do**
$\quad$ $u_1(\cdot, i) \leftarrow u_1(\cdot, i) \cdot w_x$
$\quad$ $u_2(\cdot, i) \leftarrow u_2(\cdot, i) \cdot w_x$
**end**
$\eta \sim \mathcal{N}(0_{k \times (1+v+k)}, 1)$
**for** $a \leftarrow 1$ **to** $k$ **do**
$\quad$ $u_1^-(a) \leftarrow \sum_{i=1}^{1+v+d} \mathbb{1}[u_1^-(a,i) < 0] u_1^-(a,i)$
$\quad$ $u_2^-(a) \leftarrow \sum_{i=1}^{1+v+d} \mathbb{1}[u_2^-(a,i) < 0] u_2^-(a,i)$
$\quad$ $u_1^+(a) \leftarrow \sum_{i=1}^{1+v+d} \mathbb{1}[u_1^-(a,i) > 0] u_1^-(a,i)$
$\quad$ $u_2^+(a) \leftarrow \sum_{i=1}^{1+v+d} \mathbb{1}[u_2^-(a,i) > 0] u_2^-(a,i)$
**end**

Generating distribution of actions:
$p(A = \textbf{STOP}) = p_{\textbf{STOP}}$
**for** $a, a' \in \{1, ..., k\}$ **do**
$\quad$ $\Delta(a, a') \leftarrow \|u_1(a) - u_1(a')\|_2^2 + \|u_2(a) - u_2(a')\|_2^2$
**end**
**for** $h \in \mathcal{H}$ **do**
$\quad$ $v = [1; x; o(h)]$
$\quad$ **for** $a \in \{1, ..., k\}$ **do**
$\quad\quad$ $\tilde{p}(a) \leftarrow e^{\eta(a, \cdot)^\top v}$ **for** $a' \in h$ **do**
$\quad\quad\quad$ $\tilde{p}(a) \leftarrow \tilde{p}(a) \cdot \Delta(a, a')$
$\quad\quad$ **end**
$\quad$ **end**
$\quad$ **for** $a \in \{1, ..., k\}$ **do**
$\quad\quad$ $p(A = a \mid h, A \neq \textbf{STOP}) \leftarrow \frac{\tilde{p}(a)}{\sum_{a \in \{1,...,k\}} \tilde{p}(a)}$
$\quad$ **end**
**end**

Generating distribution of potential outcomes:
**for** $x \in \mathcal{X}, z \in \mathcal{Z}$ **do**
$\quad$ **for** $a \leftarrow 1$ **to** $k$ **do**
$\quad\quad$ $v \leftarrow [1; x; z]$
$\quad\quad$ $y_0(a) \leftarrow u_1(a, \cdot)^\top v$
$\quad\quad$ $y_0(a) \leftarrow \frac{(n_y - 1)(y_0(a) - u_1^-(a))}{(u_1^+(a) - u_1^-(a))}$
$\quad\quad$ $\gamma(a) \leftarrow u_1(a, \cdot)^\top v$
$\quad\quad$ $\gamma(a) \leftarrow \frac{(\gamma(a) - u_2^-(a))}{(u_2^+(a) - u_2^-(a))}$
$\quad\quad$ **for** $y \leftarrow 1$ **to** $n_y$ **do**
$\quad\quad\quad$ $\tilde{p}(a, y) \leftarrow f_{\text{cauchy}}(y; y_0(a), \gamma(a))$
$\quad\quad$ **end**
$\quad\quad$ **for** $y \leftarrow 1$ **to** $n_y$ **do**
$\quad\quad\quad$ $p(Y(a) = y \mid x, z) \leftarrow \frac{\tilde{p}(a,y)}{\sum_{y=1}^{n_y} \tilde{p}(a,y)}$
$\quad\quad$ **end**
$\quad$ **end**
**end**

---