[Reviews · NeurIPS 2020]

Review 1

Summary and Contributions: The papers formalises and solves the the problem of assigning treatments under the knowledge of a casual graph that takes into account the effects of treatments with outcomes and the effect of potential confounders that can be observed or not. Three algorithms are proposed and evaluated: an exact solution that uses dynamic programming a greedy approximation for high dimensional settings and a model-free approach motivated by the RL literature. Apart from the new algorithms and the experimental work, the paper is full of theoretical insights that characterise the identification of the models under observed and unobserved confounders.

Strengths: - The problem in very relevant particularly in healthcare. - The paper is extremely well written, and full of interesting theoretical and practical results. - The experimental section is convincing.

Weaknesses: I don't see any major weaknesses in this work, and i think that it is a pretty solid contribution. One can always discuss whether the causal graph that the authors assume is given is realistic or not depending on the problem but in this case I think that the authors consider an scenario that is general enough

Correctness: Yes, to my knowledge the approach is correct and the code seems to be well written too.

Clarity: Yes, the paper is very clear. Some sections are a bit dense, but the authors have done a good job motivating the results and introducing the notation carefully.

Relation to Prior Work: Previous work is cited and I am not aware of any important piece of literature that is missing.

Reproducibility: Yes

Additional Feedback: I think that his is pretty solid work and I don't have any major complain or comment.


Review 2

Summary and Contributions: This paper aims at designing a method for searching for an effective medical treatment. The authors use causal inference framework to formalise the problem and provide a dynamic programming algorithm to solve the problem. The proposed method is evaluated by both synthetic and real world data sets, and a comparison has been conducted.

Strengths: The paper studies an interesting problem. The experiment is on a real world data set.

Weaknesses: I do not see how causal inference framework help formulate and solve the problem. Firstly, I am confused by the use of potential outcome Y_s (a). If the “a” is binary, it normally contains 1 or 0 representing treatment and control. We can only observe one of two potential outcomes, and the other needs to be estimated (the counterfactual potential). When reading this paper, I could not see the discussions on the observed potential outcome and counterfactual outcome, and I could not see their link to the treatment effect either. Secondly, I doubt whether Figure 1(a) is realistic. It says that the outcome Y_T is only determined by A_T. It is surprise that the most recent treatments and outcomes do not affect the current outcome. Thinking about that the A_T is set by a doctor and it a manipulation in the causal graph. Once A_T is set, the Y_T is determined by A_T only. I think that this is very simple and unrealistic in practice. Thirdly, the proposed method and its variants perform similarly to a model free RL method, which is a naïve dynamic programming based method (NDP), in the real world data set. Authors argue that the proposed methods offer more transparent trade-off between search time and treatment efficacy, but this has not been elaborated in the experiment section.

Correctness: No flaws are identified.

Clarity: The paper is well written.

Relation to Prior Work: Discussions on related work are reasonable.

Reproducibility: Yes

Additional Feedback: In causal inference, a distinction of observed outcomes and counterfactual outcomes is very important. In the paper, such distinction is not clear. Authors point to Theorem 1 to link two together. I read Section 4 again, I could not see such a distinction. So, I cannot see the connection. From authors' reply "To solve our policy optimization problem (1), it is not necessary to impute all counterfactuals. For example, in the binary case given by R2, if an action a = 0 has been tried, and Y (0) observed, only the probability that Y (1) > Y (0) is required to solve the problem.", I am even not sure if the authors understanding is correct. Each individual has two potential outcomes, Y(0) and Y(1). When Y(0) is observed for an individual, we need to estimate his/her counterfactual potential outcome Y(1). Individual treatment effect is Y(1) - Y(0), and the average treatment effect in a population is E[Y(1) - Y(0)]. If we do not estimate the counterfactual outcome for the individual, we do not know the individual treatment effect. If we do not estimate the counterfactual outcomes of all in the population, we do not have the average treatment effect. I am not sure how authors estimate the probability that Y (1) > Y (0) without imputing all counterfactuals. In this equation. Y(1) is a potential outcome, not an observed outcome. This is exactly what I would like to know how to estimate probability that Y (1) > Y (0) in observed data set.


Review 3

Summary and Contributions: Some methods are discussed to search for causally near-optimal treatments, such as the best antibiotic to use for a particular patient. A greedy approach is recommended.

Strengths: Theoretical claims are sound. I have not heard of this approach before (am I naive?), but the motivation make sense. I was particularly taken with the empirical example--this clearly indicates how this approach might be of use to practitioners.

Weaknesses: It was stated that experts are still performing better than this approach on the empirical example, which is definitely a limitation. The remark was that the experts might be privy to information that the program was not, a typical problem with trying to provide programmatic advice in place of expert advice. However, if that's the case, would it be feasible to try to include more information to the optimization to close the gap? If not, what is the scenario? In a hospital, say, does the patient's information get coded into the program, the button pushed, and an antibiotic output, to be considered by the physician? Does a ranked list of potential antibiotics get output for the physician to choose from? It's unclear to me.

Correctness: The theoretical claims, and the arguments in the appendix, look good. The empirical methodology is very well done.

Clarity: The paper is VERY WELL WRITTEN! Thanks!

Relation to Prior Work: Yes--different from previous contributions (that I know of).

Reproducibility: Yes

Additional Feedback: I appreciated the author feedback; I thought it was very clear.


Review 4

Summary and Contributions: The paper proposes a formal problem space for efficiently identifying a treatment which meets or exceeds a predefined level of efficacy. This problem space does not involve identifying a more efficaous treatment, or combination of treatments, but rather focuses on getting to an acceptably effective treatment, using an iterative process, as quickly as possible.

Strengths: The problem space is clearly defined, mathematically/statistically defensible, and may spur additional experiments to surpass the baselines that the authors have established. The manuscript itself was clearly written, easy to follow, and provided ample support for the claims that were made. The inclusion of (a) proofs, (b) synthetic data, (c) real-world observational data are laudable.

Weaknesses: The primary weaknesses of the paper do not concern the methods, rigor, or writing of the submission, but whether: a) The specific problem for which the approach was developed is meaningful AND b) whether the results for the best approach are meaningful in the context of application chosen by the authors

Correctness: Yes, claims logically flow from problem statement and results. Methodology is well supported One nit-picky question if I may: In Figure 2a, it is curious that the mean effect of the emulated doctor is so much lower than that of the only real-world data point, that of the actual doctor. This may indicate that the approach for defining the emulated doctor needs refinement and that the curve for it may be artificially low

Clarity: Clearly written paper. Perhaps a bit too much mixing of mathematical notation in the middle of sentences, but not out of reason.

Relation to Prior Work: The current work was clearly linked to prior contributions across multiple fields.

Reproducibility: Yes

Additional Feedback: Overall, this is a very clear and well supported manuscript. However, given this work is being presented with specific healthcare applications in mind, it's challenging to separate the clinically unimpactful results from the rigorous definition of the problem space and methodology. The results clearly demonstrate that the authors approach is superior to clinical baselines and alternative approaches for the application addressed in this paper. However, the improvement offered is minuscule. Additionally, the authors state (and I agree) that the approach may not be applicable for the great majority of clinical use cases involving observational data. Finally, the authors also acknowledge that policies which minimize search time may lead to sub-optimal short-term response. This may be critical in many applications, including the antibiotic use case that the authors presented. Short term response, even if insufficient in the long term, may often times keep a patient alive long enough to find a more optimal solution. Getting to the finish line faster is not very exciting if the patient has died half way through the race. In summary, the paper is excellent but the impact is minimal.

[Author Response · NeurIPS 2020]

We thank the reviewers for their excellent feedback and for recognizing the value of our work to the NeurIPS audience.

**R1, R2: Realism of the causal graph.** In our model, the outcome $Y_t$ is determined not only by $A_t$, but also by the
unobserved moderating variable $Z$ and possibly exogenous variables. $Z$ may capture, for example, the disease state
of a subject and introduce *correlations* between outcomes $Y_s$ and $Y_t$ for $s < t$. It is a common working assumption
that previous actions and outcomes, $A_s, Y_s$ for $s < t$, do not have direct *causal* effect on $Y_t$ when $Y_t$ represents the
symptoms of chronic conditions where drugs do not affect the underlying disease state; given sufficient time between
treatments, symptoms return to a baseline level until another treatment is started. In addition to rheumatoid arthritis [1],
which is used as motivation in the manuscript, examples include depression [2] and Parkinson's disease [3].

**R2: Usefulness of causal framework.** The causal framework is necessary for distinguishing the observational
outcome $Y_t$ from the potential outcome $Y(a)$. Only under certain assumptions can $Y(a)$ be estimated from $Y_t$.
Sufficient conditions for this are given in Section 4. The distinction is particularly important in the case where $Y(a)$ is
not fully identifiable from observational data due to unobserved confounding, as discussed in Section 4.2.

**R2: Observed and counterfactual outcomes.** The relationship between counterfactual outcomes $Y(a)$ and observed
outcomes $Y_t$ is established in Theorem 1. The LHS of (4), $\rho(h_s)$, is a function of counterfactual outcomes, as defined
in (2), and the RHS is a function of observed outcomes $Y_t$. As remarked after Theorem 1, under our assumptions, a
model of $p(Y_t \mid H_t = h, A_t = a)$ is sufficient for estimating the distribution of $Y(a)$. To solve our policy optimization
problem (1), it is not necessary to impute all counterfactuals. For example, in the binary case given by R2, if an action
$a = 0$ has been tried, and $Y(0)$ observed, only the probability that $Y(1) > Y(0)$ is required to solve the problem.

**R2: Relation to model-free RL.** R2 is correct that the model-free method NDP compares similarly to the other
methods in the antibiotics experiment. However, we show in Figure 1b that the qualitative behavior of NDP as a function
of dataset size is very different from that of CDP and CG. Additionally, as shown in Appendix A.7 (Thm A7), NDP is
suboptimal in the general case. We hope that these contributions are recognized. By a "transparent" tradeoff, we refer to
the meaning of the parameters $\delta$ (CDP, CG) and $\lambda$ (NDP). $\delta$ is directly interpretable as a probability threshold at which
we are satisfied with the best-so-far treatment (as used in the antibiotics experiment). The value of $\lambda$ does not have an
immediate interpretation as a level of certainty of near-optimality—*the tradeoff for a fixed $\lambda$ varies across datasets*.

**R3, R4: Comparison with experts and the emulated expert.** R3 is correct that it is feasible in practice to include
more information in the policy so that it compares more favorably to experts in the first step. Similarly, the accuracy of
the emulated doctor, remarked upon by R4, could be improved by using more information to emulate the doctor policy.
In our experiment, we intentionally kept the patient representation small because the number of samples was fairly
limited. The emulated expert in our study attempts to approximate the expert's policy with the same information given
to the other algorithms. As such, it serves as an *imitation learning* baseline to complement the policy optimization
approaches developed in this work. We will clarify this choice in the paper. We note, however, that our algorithm is
trying to achieve a different goal than the expert. While experts may attempt to prescribe the best action on the first try,
we try to minimize the expected number of tested treatments. In that sense, the expert can be thought of as a greedy
agent, which our paper argues is not always optimal, if we're trying to minimize needless trial and error on patients.

**R4: Impact of this work.** We certainly agree that our method is not suitable for all applications. However, there is a
large class of medical conditions and treatments which fall exactly under the specifications of our model. In fact, our
motivation for this work is the result of working with active clinicians treating rheumatoid arthritis. The application
of our method for this purpose is an ongoing project which will be aimed at the clinical research community, but we
appreciate the reviewer's feedback which highlighted that the use cases of our method are not readily apparent from
reading the paper. We will add a description of the RA problem to the paper to make it more concrete and demonstrate
a motivating use case, along with a discussion of other uses such as in treating psychiatric disorders. Finally, we believe
that the problem has applications also outside of medicine, such as for general recommendation systems.

**R4: Short-term response.** It is true that short-term response is critical for some applications and should not be
discounted; this is a potential challenge also for reinforcement learning which optimizes long-term return, possibly
sacrificing immediate rewards. Our goal is to find a near-optimal treatment in as few steps as possible which is an
important consideration in other applications [2]. If an optimal treatment *can* be reliably identified in a single step,
the algorithm is incentivized to do so. Short-term success is sacrificed only if there is great uncertainty about which
treatment is likely to work and this can be reduced by a sub-optimal treatment. Our greedy approximation incentivizes
short-term response by preferring actions higher that are likely to have a higher outcome (see 5.2). Such incentives
could be incorporated into the dynamic programming solution as well, and is an interesting direction for future work.

[1] J. R. O'Dell. Therapeutic strategies for rheumatoid arthritis. *New England Journal of Medicine*, 350(25):2591–2602, 2004.
[2] M. F. Pradier, T. H. McCoy Jr, M. Hughes, R. H. Perlis, and F. Doshi-Velez. Predicting treatment dropout after antidepressant initiation. *Translational psychiatry*,
10(1):1–8, 2020.
[3] M. Stacy, A. Bowron, M. Guttman, R. Hauser, K. Hughes, J. P. Larsen, P. LeWitt, W. Oertel, N. Quinn, K. Sethi, et al. Identification of motor and nonmotor
wearing-off in parkinson's disease: comparison of a patient questionnaire versus a clinician assessment. *Movement disorders: official journal of the Movement
Disorder Society*, 20(6):726–733, 2005.


[Meta-Review · NeurIPS 2020]

The papers considers the problem of assigning treatments with the side information about the casual graph. The authors propose 3 algorithms and evaluate their performance: an exact algorithm based on dynamic programming, a greedy approximation for high dimensional settings, and a model-free approach motivated by the RL literature. The reviewers for the most part, felt that the application of knowledge of the causal graph to RL is novel and interesting. They were satisfied with the theoretical insights as well as algorithmic contributions and experimental work.